# In situ detection of the protein corona in complex environments

Monica Carril [iD] [1,2], Daniel Padro[1], Pablo del Pino[1,3,4], Carolina Carrillo-Carrion[1], Marta Gallego[1]
& Wolfgang J. Parak [iD] [1,3,5]

Colloidal nanoparticles (NPs) are a versatile potential platform for in vivo nanomedicine. Inside blood circulation, NPs may undergo drastic changes, such as by formation of a protein corona. The in vivo corona cannot be completely emulated by the corona formed in blood. Thus, in situ detection in complex media, and ultimately in vivo, is required. Here we present a methodology for determining protein corona formation in complex media. NPs are labeled with $^{19}$F and their diffusion coefficient measured using $^{19}$F diffusion-ordered nuclear magnetic resonance (NMR) spectroscopy. $^{19}$F diffusion NMR measurements of hydrodynamic radii allow for in situ characterization of NPs in complex environments by quantification of protein adsorption to the surface of NPs, as determined by increase in hydrodynamic radius. The methodology is not optics based, and thus can be used in turbid environments, as in the presence of cells.

[1] CIC biomaGUNE, San Sebastian 20014, Spain. [2] Ikerbasque, Basque Foundation for Science, Bilbao 48011, Spain. [3] Fachbereich Physik, Philipps Universität Marburg, Marburg 35037, Germany. [4] Centro Singular de Investigacion en Química Biolóxica e Materiais Moleculares (CIQUS), and Departamento de Física de Partículas, Universidade de Santiago de Compostela, Santiago de Compostela 15782, Spain. [5] Fachbereich Physik and CHyN, Universität Hamburg, Hamburg 20355, Germany. Correspondence and requests for materials should be addressed to M.C. (email: mcarril@cicbiomagune.es) or to W.J.P. (email: wparak@cicbiomagune.es)

Colloidal nanoparticles (NPs) are a versatile potential platform for in vivo nanomedicine[1]. Inside the blood circulation, the NPs may undergo drastic changes, e.g., by the formation of a protein corona[2]. The dynamic development of protein corona formation has been studied extensively by mass spectroscopy or other techniques[3]. Such measurements have been performed in relevant solutions such as blood[4], which, however, requires extraction of the NPs and removal of unbound excess proteins, leading to a loss of the equilibrium properties[5]. Alternatively, protein corona formation can be observed based on measuring the associated increase in the NPs' hydrodynamic radius and reduction in diffusion coefficient[5]. In case only protein–NP complexes, but no free proteins, are subject to diffusion coefficient measurements, protein corona formation can be quantified in situ, without removal of excess proteins[6,7]. Several optical methods, such as fluorescence correlation spectroscopy (FCS)[6] or depolarized dynamic light scattering (DDLS)[7], have been established for in situ quantification based on diffusion coefficient measurements. However, in complex media such as blood or even tissue, optical detection suffers from light scattering. Unfortunately, the in vivo corona cannot be completely emulated by the corona formed in blood, etc[8]. Thus, in situ detection in complex media, i.e., ultimately in vivo, is required.

In this work, we present a non-optical methodology for determining protein corona formation in complex media. NPs are labeled with $^{19}F$ and their diffusion coefficient is measured using $^{19}F$ diffusion nuclear magnetic resonance (NMR) spectroscopy, recording a diffusion-ordered nuclear magnetic resonance spectroscopy (DOSY) experiment[9]. NMR has long been used for structural studies of proteins[10,11]; however, herein we propose the use of $^{19}F$ diffusion NMR to observe changes in hydrodynamic radius of NPs upon adsorption of proteins in solution and also in complex media such as blood.

## Results

**Synthesis and characterization of fluorinated NPs.** Three types of Au NPs labeled with fluorinated polyethylene glycol (PEG) ligands were first synthesized. Two of them contained additional PEG chains bearing either –COOH or –NH₂ head groups and the third type was further coated with the polymer poly(isobutylene–*alt*-maleic anhydride) (PMA), which in water also presents –COOH groups at its surface. These three types of NPs are in the following referred to as NP-F/COOH, NP-F/NH₂, and NP-F/NH₂@PMA and are depicted in Fig. 1a. Transmission electron microscopy (TEM) analysis revealed the following similar radii $r_c$ of the gold cores: $r_c = 1.5 \pm 0.7$ nm (NP-F/COOH); $r_c = 1.7 \pm 0.6$ nm (NP-F/NH₂); and $r_c = 1.7 \pm 0.6$ nm (NP-F/NH₂@PMA). All data are presented as mean value ± standard deviation. NP concentrations were determined by UV/Vis absorption spectroscopy and inductively coupled plasma mass spectrometry (ICP-MS). The NPs were characterized to possess good colloidal stability and zeta potential measurements in water-indicated negative or positive surface charge of the NPs terminated with –COOH or –NH₂, respectively: $\zeta = -13.8 \pm 1.1$ mV (NP-F/COOH); $\zeta = +5.7 \pm 0.7$ mV (NP-F/NH₂); and $\zeta = -36.2 \pm 1.2$ mV (NP-F/NH₂@PMA). All three types of NPs allowed for recording $^{19}F$-NMR signal in the form of a single peak.

**Size calculation based on $^{19}F$ diffusion NMR.** Using $^{19}F$ diffusion NMR measurements, the following diffusion coefficients (D) for these NPs in water were obtained: $D = (2.73 \pm 0.03) \times 10^{-11}$ m²/s (NP-F/COOH); $D = (2.86 \pm 0.04) \times 10^{-11}$ m²/s (NP-F/NH₂); and $D = (3.39 \pm 0.29) \times 10^{-11}$ m²/s (NP-F/NH₂@PMA). Based on the Stokes–Einstein relation, the following hydrodynamic radius $r_h$ in water could be derived from the diffusion coefficients,

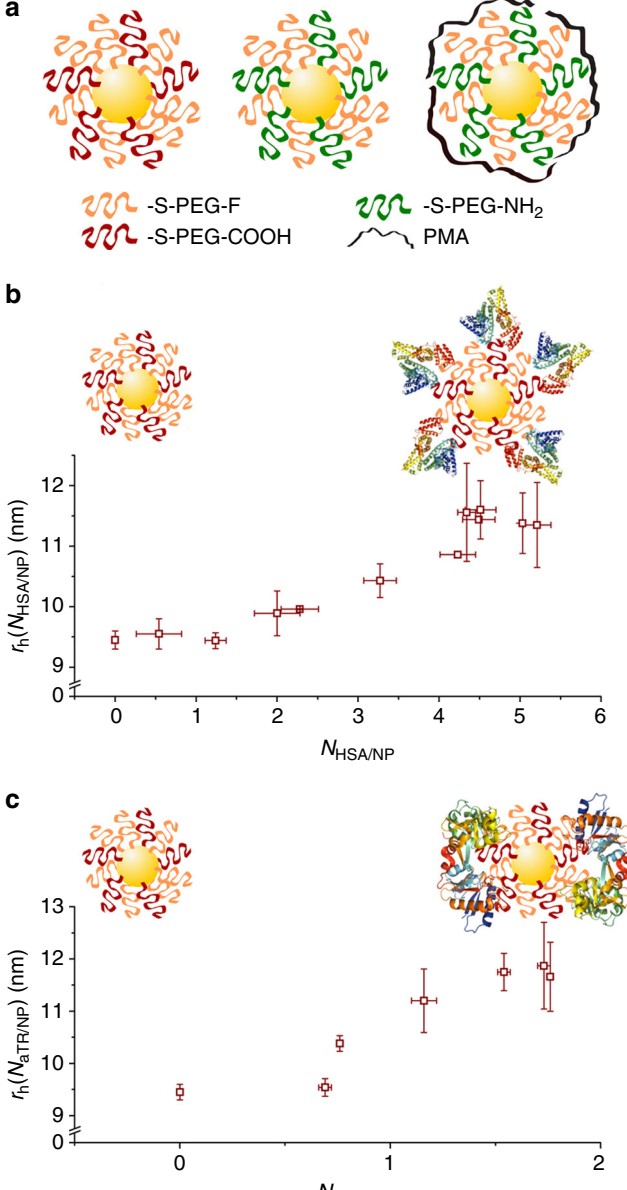

**Fig. 1** Size measurement of fluorinated NPs modified with either HSA or aTR. **a** Illustration of the three types of $^{19}F$-labeled NPs, from left to right: NP-F/COOH, NP-F/NH₂, and NP-NH₂/PMA. **b** Mean values ± standard deviation (from at least two measurements) of the hydrodynamic radii $r_h$ for NP-F/COOH covalently conjugated with increasing numbers of HSA molecules ($N_{HSA/NP}$). **c** Mean values ± standard deviation (from at least two measurements) of the hydrodynamic radii $r_h$ for NP-F/COOH covalently conjugated with increasing numbers of aTR molecules ($N_{aTR/NP}$)

similar to approaches reported in literature[12]: $r_h = 8.99 \pm 0.08$ nm (NP-F/COOH); $r_h = 8.57 \pm 0.12$ nm (NP-F/NH₂); and $r_h = 7.25 \pm 0.63$ nm (NP-F/NH₂@PMA). As expected, the hydrodynamic radii $r_h$ were a few nm larger than the core radii $r_c$, accounting for the size of the hydrated ligand shell, which is not visible in TEM[13]. This confirmed the general possibility of measuring hydrodynamic radii of $^{19}F$-labeled NPs with $^{19}F$ diffusion NMR. The obtained hydrodynamic radii were confirmed by DLS (see Supplementary Tables 15 and 16), and are also in agreement with data obtained with FCS with similar (fluorescence-labeled) NPs. Similar NPs with PEG or PMA surface coating have also been demonstrated to be highly colloidally stable, also in the presence

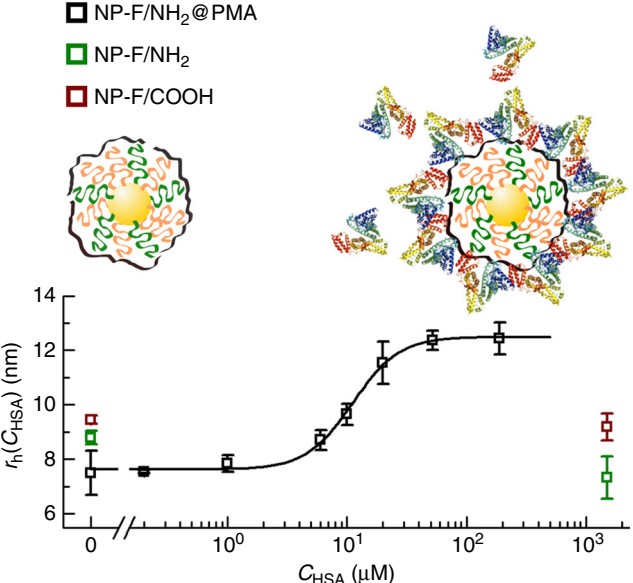

**Fig. 2** Size increase in the presence of HSA. Hydrodynamic radii $r_h \pm$ standard deviation (from at least three measurements) as measured in situ (i.e., under equilibrium with excess proteins present in solution) for the three types of NPs in the presence of increasing concentrations $c_{HSA}$ of HSA in PBS, and the corresponding fit based on the Hill model for the case of NP-F/NH$_2$@PMA, which was the only NP type that underwent an increase of size due to protein adsorption. In the case of NP-F/NH$_2$ and NP-F/COOH, no protein adsorption in terms of no significant change in hydrodynamic radius was observed

of NaCl and proteins[14]. In addition, NPs with different surface chemistry and larger overall radius were synthesized as control and thoroughly characterized, and their hydrodynamic radii were determined with $^{19}$F diffusion NMR (see Supplementary Tables 23 and 24). Results indicate that the here proposed method is stable in case the $^{19}$F labels are introduced without PEG, and theoretically for NPs with hydrodynamic radii as large as 100 nm, as long as sufficient $^{19}$F-NMR signal is detectable. However, our measurements are highly dependent on the $T_2$ values and concentration of fluorine, which will certainly limit the diffusion coefficients ($D$) measurement for nanomaterials of such large size.

**Protein corona study based on $^{19}$F diffusion NMR.** As a next step, protein adsorption to the different NPs based on increase in their hydrodynamic radii was probed by $^{19}$F diffusion NMR, using protocols similar to those used with FCS in previous reports[6]. As starting point, artificial protein coronas were generated by covalent linkage of human serum albumin (HSA) or human apo-transferrin (aTR) to the surface of NP-F/COOH through N-(3-Dimethylaminopropyl)-N′-ethylcarbodiimide hydrochloride (EDC) chemistry. The amount of attached proteins was hereby controlled by the EDC concentration. For each condition, the number of bound proteins per NP ($N_{HSA/NP}$, $N_{aTR/NP}$) was determined by Bradford assay. At the maximum EDC concentration possible without interfering with the colloidal stability of the NPs $5.21 \pm 0.17$ and $1.76 \pm 0.01$ HSA and aTR protein molecules, respectively, could be bound per NP. For all conditions the hydrodynamic radii of the NP–protein complexes were determined by $^{19}$F diffusion NMR measurements. In Fig. 1b, c, the increase in hydrodynamic radii in dependence of the number of attached proteins is shown. As expected, the more proteins are attached per NP, the more the hydrodynamic radii increases, reaching saturation when the maximum possible amount of

proteins is attached per NP. Similar measurements of NP–protein size vs. number of attached proteins could be recorded with gel electrophoresis[15], but not with standard DLS, where the experimental error was higher than the actual changes in size.

These data motivated for measuring protein adsorption in situ by $^{19}$F diffusion NMR. While for DLS free proteins would interfere with analysis, gel electrophoresis separation of NP–protein complexes from free proteins would lead to loss of equilibrium conditions and desorption of proteins from the NP–protein complexes[5]. In situ detection of protein adsorption in phosphate-buffered saline (PBS) has been reported by FCS[6]. Here the same principle was carried out with $^{19}$F diffusion NMR experiments. HSA was added in different concentrations $c_{HSA}$ to NP-F/NH$_2$@PMA in PBS, and for each concentration the hydrodynamic radius $r_h$ of the NP-HSA complexes was measured in situ, i.e., without any purification of unbound HSA excess by $^{19}$F diffusion NMR. This was possible as the $^{19}$F label was attached to the NPs, and thus the proteins themselves did not provide signal. The data shown in Fig. 2 present the typical sigmoidal curve, as known from FCS measurements[6]. Using the Hill model, the maximum number of adsorbed HSA molecules per NP was determined as $N_{HSA/NP(max)} = 65.6 \pm 2.6$, HSA adsorption is cooperative with a Hill coefficient of $n = 2.3 \pm 0.2$, the apparent dissociation coefficient is $K'_D = 13.6 \pm 0.9$ μM, and under saturation conditions the HSA corona forms a monolayer with the thickness $\Delta r_h = 4.87 \pm 0.08$ nm around the NPs. These values are on the same order of magnitude as those derived from FCS measurements on PMA-coated surfaces[6]. Note that in the geometry used here, the PEGylated $^{19}$F-ligands are overcoated by PMA. In contrast, no HSA adsorption was found in the investigated range for NP-F/COOH and NP-F/NH$_2$ (Fig. 2). For these NPs, the $^{19}$F-ligands are directly exposed to the surface. FCS measurements on similar NPs have shown that PEGylation reduces protein adsorption[16]. In case of mushroom-like coating, proteins may penetrate the PEG layer, and thus still contribute to a slight increase of the hydrodynamic radius. In case of tight PEG layers, as expected in the present case due to the direct linkage of the PEG ligands via thiols, suppression of protein adsorption is expected. In addition, the fluorine head groups also could repel proteins[17]. Effects of fluorination on protein adsorption have been discussed in the literature, mainly for planar surfaces[18,19]. In additional control experiments, we could show that in fact the hydrophobicity associated with fluorine head groups had effect on the interaction with fibrinogen (see Supplementary Figs. 49 and 50). Hydrodynamic radii increase upon natural adsorption of transferring (TR) on NP-F/NH$_2$@PMA was also studied (see Supplementary Fig. 30). Taken together, our data demonstrate that $^{19}$F diffusion NMR experiments can be used for the in situ quantification of protein adsorption.

**Protein corona study in complex media with $^{19}$F diffusion NMR.** The data shown in Fig. 2 were recorded under well-defined conditions, in which besides the NPs, only one type of protein, in this case HSA, was present in solution. As there were no impurities able to scatter and attenuate light, comparable analysis can be performed also with optical interrogation, such as FCS[6], instead of NMR. We thus wanted to perform experiments in complex media, in order to highlight the possibilities of $^{19}$F diffusion NMR experiments. All three types of NPs were exposed either to whole blood, or to blood plasma, which was donated by one of the coauthors (C. C.–C.), and the resulting diffusion coefficients of the NPs were measured in situ by $^{19}$F diffusion NMR. These measurements were indeed possible due to the absence of interfering fluorinated molecules in physiological media; hence the signal detected in $^{19}$F-NMR belongs unequivocally to the NPs. The results

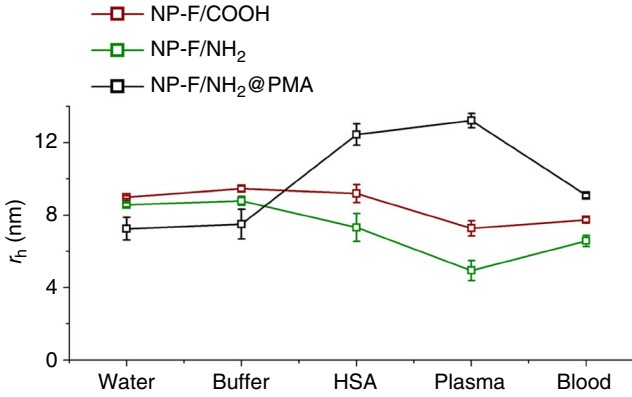

**Fig. 3** Size measurements in different media. Hydrodynamic radii $r_h \pm$ standard deviation (from at least two measurements) as measured for the three types of NPs: in water, aqueous buffer (HEPES or PBS), in the presence of HSA (under saturation conditions), in isolated plasma, and in blood

shown in Fig. 3 first prove that there is no agglomeration of the NPs either in blood or in plasma. As no drastic increase in size was observed, we can conclude that all three types of NPs remained colloidally stable and well dispersed in blood and plasma. Incubation in blood did not largely change the hydrodynamic radii of the NPs as compared to dispersion in PBS. However, in the case of NP-F/NH$_2$@PMA incubation in plasma resulted in a size increase of ca. 5.5 nm, comparable to that under saturation exposure with HSA. Thus, also in the case of plasma, there was protein adsorption, leading to a controlled thin (mono)-layer of proteins around the NPs, whereby the composition, i.e., the types of proteins, is not known. For this, NPs would have to be extracted, excess proteins removed, and the composition of the corona analyzed for example by mass spectroscopy[3]. However, the experimental finding that even in a complex mixture of proteins and other molecules as lipids there is a defined thin corona, is in agreement with data from the literature[20]. In the case of NP-F/COOH and NP-F/NH$_2$, there was some decrease of size observed in plasma, while under saturation with HSA the size of these NPs was similar to that recorded in PBS. This might be explained by shrinkage of the PEG-containing ligands in plasma, as it has been reported by other groups[12,21,22]. Data indicate that it is possible to make NPs that remain well dispersed in blood and plasma, and that $^{19}$F diffusion NMR allows for in situ analysis of the state of dispersion and protein corona formation, without the need of any purification.

As magnetic fields are not adsorbed by tissue, measurements of hydrodynamic radii of $^{19}$F-labeled NPs could also be possible in vivo. Hereby, the expected signal-to-noise ratio becomes critical. In order to obtain sufficient signal, the amount of injected NPs needs to be maximized, while still avoiding toxic side effects. Signal-to-noise ratio can also be increased by longer scanning times, which, however, would result in losing the capability of observing time-dependent changes on shorter time scales. While at the present stage, we are not able to manifest our vision with experimental in vivo data, we have estimated that injection of 200 μL of 2.7 μM NP-F/COOH solution into a mouse should allow for in situ detection of $^{19}$F signal requiring scanning times on minute time scales (see Supplementary Fig. 72). However, since in diffusion NMR, a signal intensity loss is measured, which is in addition dependent on the specific gradients applied, at this stage it is difficult to predict whether this NP-F/COOH concentration would be enough or not. Further improvement could be achieved by increasing the $^{19}$F density on the surface of the NPs and/or the fluorinated probe concentration.

After in vivo administration, NPs would circulate in the blood flow, surrounded by the cellular walls of the blood vessels. For most applications, long circulation times are desired. We emulated this scenario by adding NP-F/COOH to suspensions of THP-1 cells. Due to their PEG coating, the NPs are not supposed to be strongly incorporated by cells[23], i.e., these NPs would be expected to have decent circulation times. The hydrodynamic radii of the NPs above cells could in fact be measured by NMR and led to similar results as obtained on plain NP solutions (see Supplementary Fig. 69).

## Discussion

Still, interpretation of future in vivo results may turn out complex. In case NPs are endocytosed by macrophages (which could be partly prevented by PEGylation), they would be confined to endosomes/lysosomes, which in fact could affect diffusion measurements. After intravenous injection in situ characterization of NPs directly in vivo could result in the following scenarios. Agglomeration of NPs could be detected as drastic increase in their effective hydrodynamic radii. Degradation of NPs, in particular partial loss of the surface coating[24], would lead to substantial decrease in hydrodynamic radii. Last but not least, adsorption of proteins would be observed as small increase in NP size, depending on the type of NP used. Our methodology is merely based on measuring diffusion constants to obtain hydrodynamic radii, and is thus insensitive to information on a molecular level. While interpretation of change of hydrodynamic radii in vivo will not be straightforward as outlined in the different scenarios above, still these measurements potentially set the pavement for future in situ monitoring of the geometric properties of NPs in vivo.

$^{19}$F diffusion NMR-mediated measurements of hydrodynamic radii allow for in situ characterization of NPs in complex environments. This has been demonstrated with the quantification of protein adsorption to the surface of NPs, as determined by the increase in hydrodynamic radius. However, for soft coatings, such as PEG, protein interactions may be difficult to assess due to the coating compressibility, which may not lead to $r_h$ increase. The methodology is not optics based, and thus can also be used in turbid environment, as for example under the presence of cells. While several obstacles need to be solved, the presented methodology in principle may provide an opportunity to measure hydrodynamic radii of NPs in vivo. Realizing this vision would be an important step toward understanding what is going on in vivo after NPs have been administered.

## Methods

**Fluorine-labeled NP synthesis**. Fluorinated PEG (3 kDa) ligands were prepared by simple synthetic transformations from the corresponding commercially available thiol-protected hydroxyl-ending PEG compound. First, PEG starting material was mesylated and further substituted by perfluorinated sodium *tert*-butoxide. After acidic deprotection in the presence of trifluoroacetic acid, the so-obtained thiol ligands were used to prepare NP-F/COOH and NP-F/NH$_2$ by combining them with commercially available carboxyl and amino functionalized PEG ligands in the presence of HAuCl$_4$ and NaBH$_4$ as reducing agent and in dichloromethane as the main solvent. Once the organic solvent was evaporated, NPs were resuspended in water and purified by ultracentrifugation at $1.5 \times 10^5$ g to remove unbound ligands and very small NPs. Aggregates and bigger NPs were also removed by further centrifugation at $5 \times 10^4$ g. Subsequently, NP-F/NH$_2$ was further reacted with PMA in a mixture of dichloromethane and tetrahydrofuran overnight. Excess of PMA was removed by several cycles of centrifugation at $2 \times 10^3$ g and further treated with NaOH (aq.) to obtain NP-F/NH$_2$@PMA. All three types of NPs were characterized by TEM, UV-Vis, $^{19}$F-NMR, and Zeta potential. Full synthetic details and characterization of ligands and NPs is provided in the Supplementary Methods.

**NP diffusion measurement by $^{19}$F-NMR**. All diffusion experiments were performed in 5 mm standard NMR tubes filled with a minimum sample volume of 465 μL and a coaxial insert carrying 100 μL of a TFA solution in D$_2$O (0.024% v/v). All glassware were rinsed three times with the corresponding buffer used for each sample (HEPES or PBS) before sample loading. For the measurements of natural adsorption of HSA, all glassware were passivated with a solution of HSA (10 mg/mL) for 15–30 min and then rinsed three times with PBS. All NMR data were collected on a

Bruker AVANCE III NMR spectrometer (11.7 T, 470.59 MHz for $^{19}$F) equipped with a 5 mm $^1$H/$^{19}$F BBI probe with actively shielded z-gradient that was used in combination with a Bruker gradient amplifier providing a maximum current of 10 A, which results in a 65 G/cm gradient. $^{19}$F diffusion NMR measurements were performed using stimulated echo with bipolar gradient pulses from Bruker's sequence library (stebpgp1s) with the following parameters: 4k acquisition points, SW 15 ppm, NS ≥ 480, DS 32, D1 2 s, D20 (little delta) 0.5 s, P30 (big Delta/2) 1.5 ms, and 12 equally spaced gradient strengths from 5 to 95%. NMR measurements were performed using deuterium lock and while the sample was spinning.

**Calculation of $r_h$ from $^{19}$F diffusion NMR measurements**. Diffusion constants ($D$) for each sample were calculated by fitting the NMR signal intensity ($I$) decay in the diffusion $^{19}$F-NMR spectra to a mono-exponential decay (Eq. (1)) with a scaling factor $A$:

$$I = A \times e^{-D \cdot Z} \tag{1}$$

where $Z$ is the gradient strengths used in the measurement.

$D$ values were obtained in this manner as the average of at least two measurements. The hydrodynamic radius $r_h$ for each NP sample was calculated using the so-obtained diffusion constant $D$ and applying the Einstein–Stokes relation (Eq. (2)), assuming spherical shape for NPs:

$$r_h = k_B T / (6\pi\eta D) \tag{2}$$

where $\eta$ is the dynamic viscosity, $T$ is the absolute temperature, and $k_B$ is the Boltzmann constant.

**Data availability**. All data are available from the authors on reasonable request.

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

## Acknowledgments

This work was supported by the German Research Foundation (DFG grant DFG Grant PA 794/25-1 to W.J.P.). Parts of this work were funded by MINECO (CTQ2015-68413-R to M.C.). M.C. acknowledges Ikerbasque for a Research Fellow position. C.C.-C. acknowledges MINECO for a Juan de la Cierva—Incorporación contract. P.d.P. acknowledges financial support from MINECO (RYC-2014–16962), the Xunta de Galicia (Centro singular de investigación de Galicia accreditation 2016–2019, ED431G/09), and the European Union (European Regional Development Fund—ERDF). The authors acknowledge technical assistance by Javier Calvo for high-resolution mass spectroscopy measurements and Karsten Kantner for ICP-MS measurements.

## Author contributions

M.C., P.d.P. and C.C.C. prepared and characterized the fluorinated NPs and did data analysis and diffusion measurements. D.P. optimized diffusion protocols and did diffusion measurements. M.G. purified and characterized the fluorinated NPs. W.J.P. conceived the idea and designed the research. All authors contributed to results, discussion, and manuscript writing.

## Additional information

**Competing interests:** The authors declare no competing financial interests.

