## [Peer Review File · Nature Communications]

Reviewers' comments:

Reviewer #1 (Remarks to the Author):

It is now widely accepted that the interaction between nanomaterials and natural biomolecules within physiological biofluids is a major determinant of nanomaterial fate in vivo. Understanding and controlling nanoparticle-biomolecule interactions is thus critical to the development of safe and effective nanomaterial formulations. Owing to the complexity of physiological systems, the study of nanoparticle-biomolecule interactions has typically taken place in a simplified in vitro context. While much has been learned, it remains largely unclear to what extent in vitro results recapitulate the in vivo reality. In this study, the authors present an innovative and brilliant approach to characterize nanoparticle-biomolecule interactions within complex biological environments in situ – and potentially in vivo. While the creativity of the study is clear, several key experiments and discussion points (outlined below) are missing. A revised manuscript that addresses these issues would certainly be worth re-considering for publication in Nature Communications. In its current form it is best suited for a specialized journal.

Specific Issues

#1) The diffusion NMR strategy presented in this study is the first technique with the potential capability of characterizing biomolecule adsorption to nanoparticles in vivo in real time. While the authors did clearly identify the applicability of their technique to the in vivo case, they stopped short of actually performing the relevant experiments. While this is understandable given the additional effort, facilities, and expertise required, such experiments would conclusively demonstrate the applicability of this method to the real-time in vivo study of nanoparticle-biomolecule interactions. Conclusive demonstration of such a capability would represent a major advance in the field.

#2) For benchmarking, the hydrodynamic radii of the nanoparticles under study determined by diffusion NMR should be compared to the hydrodynamic radii determined using other standard techniques, in particular dynamic light scattering (DLS) or fluorescence correlation spectroscopy (FCS). This analysis should be performed in water, relevant buffers (PBS), and simplified biomolecular mixtures.

#3) All of the gold nanoparticle formulations used in this study bear a poly(ethylene glycol) (PEG) surface coating. PEG is typically applied to eliminate unwanted biomolecular interactions. Indeed, it is likely that the PEG coating is responsible for the minimal increase in hydrodynamic radius observed by the authors in blood and plasma. However, this is a special case and not generally applicable for all nanoparticle formulations. For comparison, it would be useful to prepare an additional ¹⁹F-labeled nanoparticle formulation that interacts strongly with biomolecules to demonstrate that the NMR-based technique can be used to effectively characterize the formation of a thick biomolecular corona in plasma and/or blood.

#4) The authors state that “there is no agglomeration of the NPs either in blood nor in plasma as no drastic increase in size was observed.” It may indeed be the case that the particles do not aggregate extensively. However, it is possible that a smaller fraction of the particles form lower-order aggregates (dimers, trimers, etc...) that only slightly skews the effective hydrodynamic radius determined by the Stokes-Einstein equation. Would the authors discuss this possibility in relation to the form of the NMR signal intensity decay? Moreover, there is certainly precedent in the literature for otherwise well-dispersed nanoparticles aggregating extensively upon contact with a biofluid. An additional general discussion of how one might de-convolve an increase in Rh due to aggregation from an increase in Rh due to the deposition of biomolecules is necessary.

#5) How does the ^{19}F -labeling influence biomolecule adsorption? While this may be difficult to interrogate directly, it could be studied indirectly by varying the density of the ^{19}F labels and observing how this changes the resulting R_h in plasma and/or whole blood.

#6) Would the authors discuss why the $\text{F}/\text{NH}_2@PMA$ coated particles show a significantly larger hydrodynamic radius in plasma than in blood? And why, in contrast, the F/COOH and F/NH_2 show a smaller hydrodynamic radius in plasma than in blood? These are surprising findings and warrant a more thorough discussion than is currently given in the text.

#7) It would be helpful if the authors discussed the applicability of the ^{19}F -labeling strategy to other nanoparticle systems beyond PEG-coated gold nanoparticles. Furthermore, briefly outlining a general strategy to label a nanoparticle of interest would benefit researchers attempting to apply this technique in their studies.

#8) Consider shortening the abstract by at least 50% by eliminating non-essential statements. This will improve the approachability of the study.

Reviewer #2 (Remarks to the Author):

Carril et al. present an interesting study which may have some impact in the field. Their idea to use ^{19}F diffusion NMR to study nanoparticle – protein interactions are novel and may prove useful to get a better understanding for how some nanoparticles behave in complex biological fluids. Overall, it is an impressive amount of work the authors have put together. However, there are some questions and remarks that need to be addressed before I can recommend it to be published.

To start with I have to say that it is nice to see an attempt to use NMR in the protein nanoparticle research field, however, I lack some knowledge to earlier attempts to use NMR within the field like Engel et al. and Lundqvist et al.

In general I lack information and discussions about the drawbacks and limitation of the method. For example, the authors use ^{19}F labelled PEG that they then bind to the particle surface to be able to detect the particle with NMR. This raises a couple of questions.

- The relaxation of ^{19}F . Can you still use ^{19}F if you bind it directly to the NP surface (with the current set up, with a long linker, I guess it relax in a different way than if it was bound to the surface or if a much shorter linker was used).

- What are the size limits of NPs that can be study with ^{19}F diffusion in the case of using PEG and in the case of having it at or close to the NP surface?

- Why use PEG? PEG is normally used to limit the protein interaction with a material surface. With a PEGylated surface, we actually do not learn anything about how the NP (without PEG) would interact with its surrounding when mixed with a biological solution.

- I find the derived NP sizes intriguing. The core of the $\text{NP-F}/\text{NH}_2@PMA$ is the biggest, while after being covered with PEG and additional PMA the particle is reported to be the smallest. Is this caused by less PEG, more ordered structure due to the PMA layer, a change in the water layer or a combination?

- When talking about sizes. I interpret the reported values as radii, however, the authors refer to hydrodynamic diameter r_h which may be a little confusing.

- Coming back to the question about NPs sizes. It is only NP-F/NH₂@PMA that according to the diffusion measurements increases in size when mixed with HSA, plasma and blood. The other two particles actually decrease in size. The authors say that it could be because the shrinkage of the PEG-containing ligands in plasma and have a ref for this. However, I do not find anything about this in the reference. I find that they compare brush and mushroom models for the PEG layer, saying that with a mushroom coverage with PEG there may still be available NP surface area for proteins to bind to. However, let's not bother about that. Instead, I would ask the authors to speculate a little about the cause of the shrinking particles. Postulate that the answer to the question above is that PMA restrict the PEG and make it tighter connected to the surface. Would it then be too far stretch to think that proteins may do the same thing? With that assumption it would be possible to speculate that HSA does not bind to (the PEG) NP-F/COOH but may bind (or at least interact, depends on your definitions) to the NP-F/NH₂. However, both plasma and blood contains biological macromolecules that binds to both types of PEG but they composition may differ in plasma compared to blood.

Some comments for the excellent SI.

A general thing. The texts inside the figures look like a wave in my version.

- Scales for the NMR spectrum. In the current version it is really hard to see the low intensity signals. Please change the y-axis so a reader actually can see the signals and make a judgment of impurities.

Side 8. Would it not be better to talk about force when you report r_{cf} ?

- I normally acknowledge ImageJ with a reference.

- Why is it so much fewer NP-F/NH₂@PMA analyzed from the TEM micrographs?

- Table S.II-2. What happened with NP_HSA_2_cov?

- Section V. Did any particle stick in the clot?

Reviewer #3 (Remarks to the Author):

Review on « In situ detection of the protein corona in complex environments » by Monica Carril, Daniel Padro, Pablo del Pino, Carolina Carrillo-Carrion, Marta Gallego, Wolfgang J. Parak submitted for publication in Nature Communications.

The manuscript submitted by Carril et al. introduces the use of ¹⁹F DOSY NMR to determine the diffusion coefficient of nanoparticles (NPs) in complex environment containing proteins and study the formation of a protein corona around the particles.

If the introduced methodology is particularly elegant some weaknesses in the manuscript makes it not suitable for publication in Nature Communication:

- The gold particles used are particularly small (gold core of approximately 1.5 nm) in order to make

them compatible with the diffusion measurement (larger particles would pose problem due to the increased rotational correlation time leading to a broadening of the organic ligands signals and would request specific probes for diffusion measurement). Such small particles are usually not ideal for biomedical applications (weak plasmon band – see Figure S.I-8, no magnetic properties, ...) and larger particles are usually used. **The authors should at least give an estimate of the maximal size of particles compatible with their methodology and discuss its limitations.**

- Mixtures of thiolated ligands are grafted on GNPs. Their ratio at the GNP surface is assumed to be the same as in solution. However, grafting of different molecules using thiolated ligands can difficultly be controlled. ¹H NMR spectroscopy should allow an easy characterization of the grafting densities of the different ligands on the NPs, which could influence the adsorption of proteins. TGA would also be a more robust method to determine the fraction of organic ligands compared to ICP-MS. **The characterization of the functionalized NPs should be improved.**

- ¹⁹F DOSY experiments are performed under spinning. Contradictory indications are given in the literature regarding the spinning or not of the samples during DOSY experiments. However, as reported in publications by the Gareth Morris group: “The problems with spinning are that it generates mechanical instabilities, and that unless scrupulous care is taken to synchronise the timing of gradient pulses with the sample rotation, small spatial inhomogeneities in the gradient field will lead to large signal disturbances. For these reasons sample spinning is not recommended for accurate diffusion measurements. »(see <http://dx.doi.org/10.1016/j.jmr.2014.12.006>)
As the differences in the diffusion coefficients measured in different sets of conditions are quite small, have these precautions be taken in order to ensure that experiments are run in the best conditions? Δ (“big delta”) should be mentioned as diffusion delay and δ (“little delta”) as the gradient length.

- Colloidal suspensions are difficult to stabilize in physiological conditions. Was the diffusion coefficient of the NP suspensions characterized in different ionic strength conditions in order to ensure that the - often weak - observed effects are not induced by a change in the IS of the suspension?

- The measurement, based on ¹⁹F NMR DOSY experiments, is smart but restricted to fluorinated ligands. To which extent could this layer of F atoms exposed to the solvent modify the NP/protein interactions (as suggested from Figure 2)? Has protein adsorption been characterized with fluorinated vs hydrogenated ligands using more conventional methods to highlight potential differences?

- The developed method is compared to FCS and DLS to quantify in situ formation of a protein corona around NPs. However, these advantages are not clearly described and the limitations are not discussed:

1) *Sensitivity of the NMR technique*: the whole fraction of ¹⁹F nuclei contained in a 2ml sample is requested to perform a measurement in reasonable time. How could this be extrapolated to particles dispersed in tissues or injected in animals, as the particles, distributed in these complex matrices, will experience different environments and diffusion coefficients (in addition to the fact that blood flow is necessary in a living organism) ?

2) *Dependence on the suspension characteristics*: sample viscosity is a crucial parameter in the Einstein-Stokes relation. Even in a media as plasma, TFA needs to be used as internal reference in order to determine the viscosity of the sample – and the 10% precision obtained is an important source of uncertainty on the calculated rh value. Should TFA also be injected into animals when performing in vivo experiments? The variations (decrease) obtained for the rh of NP-F/CO₂H and NP-F/NH₂ in blood/plasma vs water highlight this problem. Similarly, the decreased r_{h} for the NP-F/NH₂ in the presence of HSA

should be commented. **If the effect of the physico-chemical properties of the suspension (ionic strength, pH,...), or the presence of plasma components that lead to a shrinkage of the PEG layer, cannot be distinguished from the interaction with proteins, no information on the protein corona formation can be extracted from the measurements!**

In conclusion, if the introduced methodology is very elegant and allows the measurement of diffusion coefficient of NPs in complex media without interference of the protein signals (or of those of other analytes present in blood samples), its limitations and weaknesses are not sufficiently discussed and the obtained information as the potential in-vivo application seems largely overestimated.

All changes in the manuscript and supporting information have been highlighted in yellow. Apart from the suggestions of referees, we have performed the following additional changes to improve format and data of the supporting information:

- For consistency, we have renamed transferrin protein as TR instead of TF in the text and figures.
- NMR spectra in the section I of the Supporting Information have been labeled as figures with their corresponding caption.
- Some mistakes in the DLS data transcription were detected and have been corrected.
- The introduction of a substantial amount of new figures, tables and sections has led to extensive renumbering.

Reviewer #1 (Remarks to the Author):

It is now widely accepted that the interaction between nanomaterials and natural biomolecules within physiological biofluids is a major determinant of nanomaterial fate in vivo. Understanding and controlling nanoparticle-biomolecule interactions is thus critical to the development of safe and effective nanomaterial formulations. Owing to the complexity of physiological systems, the study of nanoparticle-biomolecule interactions has typically taken place in a simplified in vitro context. While much has been learned, it remains largely unclear to what extent in vitro results recapitulate the in vivo reality. In this study, the authors present an innovative and brilliant approach to characterize nanoparticle-biomolecule interactions within complex biological environments in situ – and potentially in vivo. While the creativity of the study is clear, several key experiments and discussion points (outlined below) are missing. A revised manuscript that addresses these issues would certainly be worth re-considering for publication in Nature Communications. In its current form it is best suited for a specialized journal.

⇒ We thank the reviewer for his/her generally positive comments about the general idea of our study. We have tried to address the mentioned experiments and discussion points as well as possible.

Specific Issues

#1) The diffusion NMR strategy presented in this study is the first technique with the potential capability of characterizing biomolecule adsorption to nanoparticles in vivo in real time. While the authors did clearly identify the applicability of their technique to the in vivo case, they stopped short of actually performing the relevant experiments. While this is understandable given the additional effort, facilities, and expertise required, such experiments would conclusively demonstrate the applicability of this method to the real-time in vivo study of nanoparticle-biomolecule interactions. Conclusive demonstration of such a capability would represent a major advance in the field.

⇒ We agree with the referee that performing in vivo measurements would be a major advance in the field. However, it is not straightforward to go from measurements of static samples in a NMR tube to a dynamic measurement in vivo in an actual blood stream in a living

organism. In the present work we have proved that by ^{19}F -diffusion in NMR we can measure the size of nanoparticles not only in the presence of proteins but also in complex media such as blood and plasma. Since fluorine is not present in physiological environments, the use of fluorine labels on the nanoparticles avoids interference from the surrounding media. We have also demonstrated that the fluorine content in our NPs is enough to be detected in ^{19}F -MRS in a MRI scanner. All these experiments set the ground for further research in the field and in vivo measurements are obviously our next goal. We thus clearly expressed in our manuscript that data are based on test solutions, and that potential in vitro and in vivo experiments at the current state are just a vision for the future. However, we have gone one step ahead and we present two new results based on cell cultures. First, we show that when we add nanoparticles above a cell layer (which would correspond to nanoparticles in blood capillaries), that the hydrodynamic diameters of the nanoparticles still can be measured. With this emulation we demonstrate that complex environment does not interfere with measurements. Second, we embedded cells in an agarose matrix, which brings the nanoparticles closer to cells during NMR measurements. We have then also discussed in more detail which also may be the potential problems for future in vitro and in vivo measurements. We think that the density of data presented in our work is quite high and thus merits publication.

We have included discussion on these topics in the main manuscript (page 5) and a whole new section (section VII) in the Supporting Information with all details regarding experiments in the presence of cells (pages 77-80).

#2) For benchmarking, the hydrodynamic radii of the nanoparticles under study determined by diffusion NMR should be compared to the hydrodynamic radii determined using other standard techniques, in particular dynamic light scattering (DLS) or fluorescence correlation spectroscopy (FCS). This analysis should be performed in water, relevant buffers (PBS), and simplified biomolecular mixtures.

⇒ Most samples measured by ^{19}F -diffusion NMR have been measured by DLS as well as described in detail in the Supporting Information. We have added a more detailed discussion about this comparison. We could not perform FCS with the same samples, as FCS requires a fluorescence label, but our particles had a ^{19}F label. Still, assuming that the labeling does not change the surface properties to a large amount, we can compare the NMR data of this study with FCS data of previous study. We have added a more detailed discussion about the comparison in the main manuscript (pages 3 and 4).

All DLS data can be found in the Supporting information in section IV.3 (pages 37-48), but also in Figures S.VI-22-24 and Table S.VI-5.

#3) All of the gold nanoparticle formulations used in this study bear a poly(ethylene glycol) (PEG) surface coating. PEG is typically applied to eliminate unwanted biomolecular interactions. Indeed, it is likely that the PEG coating is responsible for the minimal increase in hydrodynamic radius observed by the authors in blood and plasma. However, this is a special case and not generally applicable for all nanoparticle formulations. For comparison, it would be useful to prepare an additional ^{19}F -labeled nanoparticle formulation that interacts strongly

with biomolecules to demonstrate that the NMR-based technique can be used to effectively characterize the formation of a thick biomolecular corona in plasma and/or blood.

⇒ The choice of PEG ligands is based on their solubility and flexibility. On the one hand, we chose polyethylene glycol to counterbalance the intrinsic hydrophobicity of our highly fluorinated probe. On the other hand, PEG ensures great mobility of the fluorine atoms which is desirable to have a reasonable ^{19}F -NMR signal. In addition, PEG is ubiquitous as a ligand in NP synthesis and there are many PEGylated precursors that are commercially available and simplify greatly the ligand synthesis. It is true that PEG is supposed to have stealthy behavior in the presence of proteins, although shrinkage of PEGylated formulations in plasma has been reported, pointing at some sort of interaction (*e.g.* refs. 12, 21 and 22 of the manuscript). For a potential future application in the direction of nanomedicine the PEGylated surface chemistry thus seems relevant to us.

However, we agree with the reviewer that it is interesting to see also the effect of different surface coatings. To test our measuring protocol in a system in which proteins actually adsorbed naturally, we coated our NP-F/ NH_2 with PMA, which is known by the authors to actively interact with proteins forming a corona (see ref. 6 in the manuscript). The PMA coating in NP-F/ NH_2 @PMA was confirmed by zeta potential, which changed from +5.7 mV to -36.2 mV due to the presence of abundant carboxylate groups in PMA (Figure S.I-13) and by the reduction of hydrophobicity observed with IFT measurements, where γ_m varied from 13.73 to 18.37 mN/m (Figure S.I-14). Also, indirect confirmation was obtained from the fact that for the first time in the NPs we tried, we could observe an increase of size after NP incubation with proteins ($\Delta r_h = 4.87 \pm 0.08$ nm, Figure 2 in the manuscript and Table S.IV-6), which we did not observe at all for PEGylated NPs NP-F/ COOH or NP-F/ NH_2 . In addition, the size increase observed by us (ca. 5 nm) is in the range of what it was observed for similarly PMA-coated NPs without PEG ligands and measured by FCS (ref. 6 of the manuscript), so we believe that PEG is not affecting the corona formation in the case of PMA coated NPs, *i.e.* NP-F/ NH_2 @PMA.

We have included new experiments with PMA coated NPs to show that we can use our methodology to study protein adsorption. Hence we incubated transferrin protein with both NP-F/ NH_2 @PMA and with [NP-F/ NH_2 @PMA]*2 and the results are described in sections IV (pages 30-48) and VI.3 (pages 64-75) in the Supporting Information.

#4) The authors state that “there is no agglomeration of the NPs either in blood nor in plasma as no drastic increase in size was observed.” It may indeed be the case that the particles do not aggregate extensively. However, it is possible that a smaller fraction of the particles form lower-order aggregates (dimers, trimers, etc...) that only slightly skews the effective hydrodynamic radius determined by the Stokes-Einstein equation. Would the authors discuss this possibility in relation to the form of the NMR signal intensity decay? Moreover, there is certainly precedent in the literature for otherwise well-dispersed nanoparticles aggregating extensively upon contact with a biofluid. An additional general discussion of how one might de-convolve an increase in R_h due to aggregation from an increase in R_h due to the deposition of biomolecules is necessary.

⇒ The chemical shift window for fluorine is very broad, hence fluorine signal can be very sensitive to changes in the environment and small chemical shifts may be sometimes detected. Examples of this sensitivity can be found in Barhate et al. (Organic Letters 2008, 10, 2745) or in Granqvist et al. (Journal of Organic Chemistry 2015, 80, 7961) for other systems. We have also observed this chemical shift sensitivity ourselves, for instance, between free and bound ligand HS-PEG-F (Figure S.I-16) or between NP-F/NH₂@PMA and [NP-F/NH₂@PMA]*2 as described in section VI and shown in Figure S.VI-18. In any case, to illustrate how we could handle a situation in which there is a mixture of species diffusing differently, we have carried out an experiment with a mixture of NP-F/COOH with free unbound ligand HS-PEG-F and shown how we can detect both species and how to handle the spectra to obtain D values for each spectra. In such situation, we should extract each single spectra from the stacked spectra obtained after the measurement and de-convolve the signal for each spectra as summarized in Figure S.VI-25. The worst scenario would occur when the different species have exactly the same chemical shift and de-convolution might not be possible.

New experimental data are included in the Supporting Information in a new section dealing with analysis of mixtures, Section VI.4 (pages 75-76).

#5) How does the ¹⁹F-labeling influence biomolecule adsorption? While this may be difficult to interrogate directly, it could be studied indirectly by varying the density of the ¹⁹F labels and observing how this changes the resulting Rh in plasma and/or whole blood.

⇒ We varied the amount of fluorine in NP-F/COOH from 100 % of HS-PEG-F (NP-F) to 75% (NP-F/COOH) and 50 % (NP-F/COOH(50)) and those NPs were incubated with a high concentration of HSA (1.5 mM). The behavior observed by ¹⁹F diffusion NMR was very similar, independently from the amount of fluorine on the NP. However, we did observe differences when we changed the functionalities of the non-fluorinated accompanying ligand (from amino to carboxyl groups) with respect to NP-F after incubation with HSA or Fibrinogen. Apparently, both charge and hydrophobicity may play a major role in the interaction with proteins with PEGylated NPs, as we observed by comparing IFT and zeta potential data with diffusion measurements. In all cases, we observed shrinkage of the PEGylated shell which was greater for NP-F/NH₂ in the presence of HSA and for NP-F in the presence FIB. This suggests that HSA interactions may be driven by electrostatic interactions with positively charged NP-F/NH₂ and FIB may interact preferably through hydrophobic interactions with highly hydrophobic NP-F.

New experimental data and discussion are shown in the Supporting Information, section VI.1 (pages 56-61).

#6) Would the authors discuss why the F/NH₂@PMA coated particles show a significantly larger hydrodynamic radius in plasma than in blood? And why, in contrast, the F/COOH and F/NH₂ show a smaller hydrodynamic radius in plasma than in blood? These are surprising findings and warrant a more thorough discussion than is currently given in the text.

⇒ In principle, the fact that NP-F/NH₂@PMA behaves completely different from NP-F/COOH and NP-F/NH₂ is maybe expected, since the surface of those NPs is completely different, not

only regarding the chemical structure but also the charge and the hydrophilic profiles of each NP are substantially different (Figures S.I-13-14). NP-F/NH₂@PMA is coated with a polycarboxylate polymer, highly negatively charged which has been reported to interact intensively with serum proteins (ref. 6 of the manuscript). On the contrary, NP-F/COOH and NP-F/NH₂ are PEGylated NPs, much less charged and considerably more hydrophobic than PMA coated ones, hence they may interact differently, as we have already observed. This has also been observed before (ref. 16 of the manuscript). The fact that none of those NPs behaved equally in neither blood nor plasma may be due to several factors, such as for instance dilution. Approximately, 60 % of the volume of blood is plasma. Hence for the experiments with blood we are using a somehow diluted form of plasma, whereas when we incubate directly with plasma there is no dilution since pure isolated plasma is used. In addition, the blood cells and other blood components may interfere in unknown ways in the interaction of proteins with NPs, accounting for the different results obtained in blood or in plasma for all the examples measured. We hypothesize that NP-F/NH₂@PMA size increase is most likely due to the adsorption of proteins from blood, as it has been studied before. PEGylated NPs do not experience a size increase, but a decrease, more significant in plasma than in blood or isolated HSA. We observed similar behavior when those NPs were mixed with isolated HSA or FIB (Figure S.VI-6). There are a few reports of PEGylated systems in the literature that also suffer from shrinkage when placed in plasma [ref. 12, 21 and 22 of the manuscript]. In addition, PEG is an elastic polymer which might compress or deform by the action of certain proteins on its surface, for instance by intercalation of proteins in between PEG chains (ref. 16 of the manuscript).

We have consistently clarified through the manuscript that PEG ligands in NP-F/NH₂@PMA are overcoated with polycarboxylate PMA polymer and hence, the behavior of these NPs is different from NP-F/COOH or NP-F/NH₂.

#7) It would be helpful if the authors discussed the applicability of the ¹⁹F-labeling strategy to other nanoparticle systems beyond PEG-coated gold nanoparticles. Furthermore, briefly outlining a general strategy to label a nanoparticle of interest would benefit researchers attempting to apply this technique in their studies.

⇒ As said before, PEG is a good linker for fluorine labels based on its solubility and flexibility. However and to extend the scope of the presented methodology, we have also prepared fluorinated ligands with a linker based on an alkyl chain and a tetraethylene glycol moiety, which is also a common linker for gold NP synthesis. Those non-PEGylated fluorinated ligands linkers did not afford water dispersible NPs by themselves and other strategies had to be used in order to disperse them in water, such as the use of additional solubilizing ligands or by coating them with water soluble polymers leading to the preparation of NP-GlcF/OH and NP-TEGF@PMA. We have measured by ¹⁹F diffusion NMR the hydrodynamic radii of these new NPs coated with non-PEGylated ligands functionalized with fluorine and they ranged from 3 to 12.6 nm.

The introduction of our fluorine label can be done by two approaches: (i) the one outlined in our manuscript which consists of a nucleophilic substitution on a substrate with a leaving group (a mesylate in our case) by (CF₃)₃CONa or (ii) via Mitsunobu reaction on an hydroxyl

group (ref. 17 of the supporting information). By these two strategies it is possible to functionalize different types of ligands with fluorine atoms. However, at this stage we cannot outline a general strategy to label all kinds of NPs with fluorine because by adding fluorine to NPs surface we are rapidly increasing their hydrophobicity. Hence, the balance between fluorine content and water dispersibility of the resulting NPs is very much depending on each NP type.

New experimental data regarding the synthesis and characterization of non-PEGylated NPs is described in a new section that we have included in the Supporting Information (section VI.2, pages 61-64).

#8) Consider shortening the abstract by at least 50% by eliminating non-essential statements. This will improve the approachability of the study.

⇒ We have reduced the introduction (which was here referred to as abstract).

Reviewer #2 (Remarks to the Author):

Carril et al. present an interesting study which may have some impact in the field. Their idea to use ¹⁹F diffusion NMR to study nanoparticle – protein interactions are novel and may prove useful to get a better understanding for how some nanoparticles behave in complex biological fluids. Overall, it is an impressive amount of work the authors have put together. However, there are some questions and remarks that need to be addressed before I can recommend it to be published.

To start with I have to say that it is nice to see an attempt to use NMR in the protein nanoparticle research field, however, I lack some acknowledge to earlier attempts to use NMR within the field like Engel et al. and Lundqvist et al.

⇒ We thank the referee for pointing this out. We have included the following references from those authors at the end of the first paragraph in the manuscript:

Ref. 10 of the manuscript: High-Resolution 2D ¹H–¹⁵N NMR Characterization of Persistent Structural Alterations of Proteins Induced by Interactions with Silica Nanoparticles. Martin Lundqvist,[†] Ingmar Sethson,[‡] and Bengt-Harald Jonsson*,[†] *Langmuir* **2005** *21* (13), 5974-5979

Ref. 11 of the manuscript: Calcium-dependent Homoassociation of E-cadherin by NMR Spectroscopy: Changes in Mobility, Conformation and Mapping of Contact Regions. Daniel Häussinger, Thomas Ahrens, Hans-Jürgen Sass, Olivier Pertz, Jürgen Engel, Stephan Grzesiek, *Journal of Molecular Biology*, Volume 324, Issue 4, 6 December 2002, Pages 823-839, [http://doi.org/10.1016/S0022-2836\(02\)01137-3](http://doi.org/10.1016/S0022-2836(02)01137-3).

In general I lack information and discussions about the drawbacks and limitation of the method. For example, the authors use ¹⁹F labelled PEG that they then bind to the particle surface to be able to detect the particle with NMR. This raises a couple of questions.

- The relaxation of ^{19}F . Can you still use ^{19}F if you bind it directly to the NP surface (with the current set up, with a long linker, I guess it relax in a different way than if it was bound to the surface or if a much shorter linker was used).

⇒ The use of PEG linkers warrants a certain degree of mobility for the fluorine label even if it is linked to the NP surface. To check the relaxation of fluorine we initially measured the T_2 values of fluorine in the free ligand HS-PEG-F and in NP-F/COOH, NP-F/NH₂ and NP-F/NH₂@PMA. The values obtained for the free ligand, NP-F/COOH and NP-F/NH₂ are very high and quite similar (1277, 1007 and 975 ms, respectively). The lowest T_2 value was recorded for NP-F/NH₂@PMA and it is most likely induced by the PMA coating which may hamper fluorine mobility, but the T_2 value obtained is still very high (856 ms) (Figure S.I-17). If the ligand is too short, apart from having a much shorter T_2 value, we most likely would not be able to solubilise NPs. Indeed, we have prepared other NPs with fluorinated ligands with a linker consisting of an alkyl chain of eleven carbon atoms and a tetraethylene glycol moiety which have T_2 values below 100 ms, and yet we were able to measure their diffusion constant by ^{19}F NMR (see section VI.2, page 63).

We have measured T_2 values for all species and this information has been included in the Supporting Information in sections I (page 21) and VI (pages 63 and 69).

- What are the size limits of NPs that can be study with ^{19}F diffusion in the case of using PEG and in the case of having it at or close to the NP surface?

⇒ We have calculated the smallest diffusion constant which corresponds to the biggest system that we could measure taking into account the scanning conditions we have, the NMR scanner being used and assuming we have enough signal intensity to detect the sample, independently from the fluorinated ligand being used. Therefore, the signal attenuation under the influence of field gradient pulses in a stimulated echo using bipolar gradients follows the equation: $I = \exp[-\gamma^2 \delta^2 g^2 D(\Delta - \delta/3)]$ which corresponds to equations S.III-1 and S.III-2 combined from the supplementary information. Under the assumption that a reliable measurement of D is only possible if the field gradient pulses lead to a signal attenuation of $1/e$, with typical maximum values for the field gradient amplitude ($g = 53 \text{ G/cm}$), diffusion delay ($\Delta = 0.6 \text{ s}$) and the gradient length ($\delta = 6 \text{ ms}$) gives an estimation of minimal diffusion value of $2.5 \cdot 10^{-12} \text{ m}^2 \text{ s}^{-1}$. If we introduce this value in the Einstein Stokes relation, assuming we are in a water solution at $25 \text{ }^\circ\text{C}$ we obtain a r_h value of approximately 100 nm as the maximum theoretical size we could measure in our conditions.

As a demonstration of the potential measurement of bigger size NPs we have included a new section in the Supporting Information, section VI.3 (pages 64-75) where we analyze NPs with $r_h > 20 \text{ nm}$. A comment regarding the maximum size that could be potentially measured has also been added in page 75 of the Supporting Information.

- Why use PEG? PEG is normally used to limit the protein interaction with a material surface. With a PEGylated surface, we actually do not learn anything about how the NP (without PEG) would interact with its surrounding when mixed with a biological solution.

⇒ We have already replied to this for referee 1 and we refer to our response above.

- I find the derived NP sizes intriguing. The core of the NP-F/NH₂@PMA is the biggest, while after being covered with PEG and additional PMA the particle is reported to be the smallest. Is this caused by less PEG, more ordered structure due to the PMA layer, a change in the water layer or a combination?

⇒ The core size of NP-F/NH₂@PMA is the same as NP-F/NH₂ because the former is a derivatization of the latter and it is very similar to that of NP-F/COOH, (1.7 ± 0.6 nm vs. 1.5 ± 0.7 nm). When NP-F/NH₂ is coated with PMA, we observed shrinkage of the hydrodynamic size which we believe is due to compression of the elastic PEG chains by the PMA polymer, which is also reflected in the slight reduction of T_2 value of fluorine for those NPs from 975 to 856 ms (Figure S.I-17).

- When taking about sizes. I interpret the reported values as radii, however, the authors refer to hydrodynamic diameter r_h which may be a little confusing.

⇒ Our intention was to talk about radii r_h in all cases, the use of hydrodynamic diameter is a mistake that we have corrected all through the text. We are grateful to the reviewer to have notified us about our mistake.

- Coming back to the question about NPs sizes. It is only NP-F/NH₂@PMA that according to the diffusion measurements increases in size when mixed with HSA, plasma and blood. The other two particles actually decrease in size. The authors say that it could be because the shrinkage of the PEG-containing ligands in plasma and have a ref for this. However, I do not find anything about this in the reference. I find that they compare brush and mushroom models for the PEG layer, saying that with a mushroom coverage with PEG there may still be available NP surface area for proteins to bind to.

⇒ In the reference mentioned by the referee (ref. 12 of the manuscript), in the supporting information, Table S4, they report the hydrodynamic sizes of 3 different PEGylated polymers in PBS, DMEM and plasma and they observed shrinkage in all the cases when compared to sizes measured in PBS. Indeed, they go from sizes of 5.6 – 6.3 nm in PBS to 4.2 -5 nm in plasma.

But other examples have been reported:

From: "International Journal of Pharmaceutics 307 (2006) 93–102; Opsonization, biodistribution, and pharmacokinetics of polymeric nanoparticles. Donald E. Owens III, Nicholas A. Peppas" page 98:

"This theory makes the argument that the hydrophilic and flexible nature of the surface PEG chains allows them to take on a more extended conformation when free in solution. Therefore, when opsonins and other proteins are attracted to the surface of the particle, by van derWaals and other forces, they encounter the extended surface PEG chains and begin to compress them. This compression then forces the PEG chains into a more condensed and higher energy conformation. This change in conformation creates an opposing repulsive force that, when great enough, can completely balance and/or over power the attractive force between the opsonin and the particle surface."

From: "Colloids and Surfaces B: Biointerfaces 114 (2014) 294– 300; Shrinkage of pegylated and non-pegylated liposomes in serum. Joy Wolfram, Krishna Surib,, Yong Yang, Jianliang Shen, Christian Celia, Massimo Fresta, Yuliang Zhao, Haifa Shen, Mauro Ferrari" pages 295 and 299: *"Upon contact with serum the non-pegylated and pegylated liposomes shrank in size. The pegylated liposomes exhibited a more dramatic decrease in size (16 nm in 100% FBS) in comparison to their non-pegylated counterparts (11 nm in 100% FBS)We have shown that the characteristics of liposomes can change upon exposure to FBS. The pegylated and non-pegylated formulations displayed a serum-induced size-independent and concentration-dependent reduction in size and homogeneity"*

We have added these two references to our manuscript as ref. 21 and 22.

However, let's not bother about that. Instead, I would ask the authors to speculate a little about the cause of the shrinking particles. Postulate that the answer to the question above is that PMA restrict the PEG and make it tighter connected to the surface. Would it then be too far stretch to think that proteins may do the same thing? With that assumption it would be possible to speculate that HSA does not bind to (the PEG) NP-F/COOH but may bind (or at least interact, depends on your definitions) to the NP-F/NH₂. However, both plasma and blood contains biological macromolecules that binds to both types of PEG but they composition may differ in plasma compared to blood.

⇒ We agree with the referee that protein non covalent interactions may compress the elastic PEG coating accounting for the size decrease observed, in the same way we believe PMA is doing something similar. We believe that this interaction is not only driven by non specific van der Waals interactions but also by electrostatic and hydrophobic interactions which may explain the different behavior of our PEGylated NPs when they are not coated by PMA. We have observed this for HSA and FIB but also in plasma in blood.

We have included in the Supporting Information a whole new section which compares the behavior of some of our PEGylated NPs in the presence of HSA and FIB in terms of hydrodynamic size. We have correlated these data with the fluorine content, the zeta potential and we have also completed the characterization of our NPs with IFT measurements which can shed some light on some size modifications (section VI.1, pages 56-61).

Some comments for the excellent SI.

A general thing. The texts inside the figures look like a wave in my version.

⇒ We are sorry for that. We have checked all of our original figures and we have not seen anything like that. Hence we assume that it must have been a problem with the pdf maker software.

- Scales for the NMR spectrum. In the current version it is really hard to see the low intensity signals. Please change the y-axis so a reader actually can see the signals and make a judgment of impurities.

⇒ We have repeated some of the spectra with more scans to show better signal to noise ratio in the rest of the signals which do not belong to PEG. New NMR spectra can be found in Figures S.I-1-3 in the Supporting Information.

- Would it not be better to talk about force when you report rcf?

⇒ We have changed all the RCF by g.

- I normally acknowledge ImageJ with a reference.

⇒ We have added these two references in the Supporting Information as:

Ref. 4 of the Supporting Information: NIH Image to ImageJ: 25 years of image analysis. Caroline A Schneider, Wayne S Rasband & Kevin W Eliceiri, Nature Methods 9, 671–675 (2012) doi:10.1038/nmeth.2089

Ref. 5 of the Supporting Information: Rasband, W.S., ImageJ, U. S. National Institutes of Health, Bethesda, Maryland, USA, <https://imagej.nih.gov/ij/>, 1997-2016.

- Why is it so much fewer NP-F/NH₂@PMA analyzed from the TEM micrographs?

⇒ We have repeated the staining and measured a similar amount of NPs as for the other samples, that is > 300 NPs, as stated in the caption of Figure S.I-10 of the Supporting Information.

- Table S.II-2. What happened with NP_HSA_2_cov?

⇒ There must have been a mistake in the data transcription. In any case, to ensure everything was correct we repeated that measurement and changed Table S.II-2 and Figure S.II-2 accordingly (see page 24 of the Supporting Information).

- Section V. Did any particle stick in the clot?

⇒ We used sodium citrate as anti-clotting agent, hence no clots were formed. We have added a respective comment in page 49 of the Supporting Information.

Reviewer #3 (Remarks to the Author):

Review on « In situ detection of the protein corona in complex environments » by Monica Carril, Daniel Padro, Pablo del Pino, Carolina Carrillo-Carrion, Marta Gallego, Wolfgang J. Parak submitted for publication in Nature Communications.

The manuscript submitted by Carril et al. introduces the use of ¹⁹F DOSY NMR to determine the diffusion coefficient of nanoparticles (NPs) in complex environment containing proteins and study the formation of a protein corona around the particles.

If the introduced methodology is particularly elegant some weaknesses in the manuscript makes it not suitable for publication in Nature Communication:

- The gold particles used are particularly small (gold core of approximately 1.5 nm) in order to make them compatible with the diffusion measurement (larger particles would pose problem due to the increased rotational correlation time leading to a broadening of the organic ligands signals and would request specific probes for diffusion measurement). Such small particles are usually not ideal for biomedical applications (weak plasmon band – see Figure S.I-8, no magnetic properties, ...) and larger particles are usually used. **The authors should at least give an estimate of the maximal size of particles compatible with their methodology and discuss its limitations.**

⇒ We agree that having a small core NPs and a long PEGylated ligand favors the measurement. However, we have also prepared other NPs with r_h values ranging from 3 to 20 nm with T_2 values ranging from 71 ms to 654 ms and measured their diffusion constant with the presented methodology. Regarding the applicability of this size of NPs, we agree that they do not have applications in the plasmonic field but have potential use as contrast agents for ^{19}F -MRI (see ref. 17 in the Supporting Information).

Regarding the maximum size we can measure, we have already replied to that and we copy here the response:

We have calculated the smallest diffusion constant which corresponds to the biggest system that we could measure taking into account the scanning conditions we have, the NMR scanner being used and assuming we have enough signal intensity to detect the sample. Therefore, the signal attenuation under the influence of field gradient pulses in a stimulated echo using bipolar gradients follows the equation:

$I = \exp[-\gamma^2 \delta^2 g^2 D(\Delta - \delta/3)]$ which corresponds to equation S.III-1 and S.III-2 from the supplementary information.

Under the assumption that a reliable measurement of D is only possible if the field gradient pulses lead to a signal attenuation of $1/e$, with typical maximum values for the field gradient amplitude ($g = 53 \text{ G/cm}$), diffusion delay ($\Delta = 0.6 \text{ s}$) and the gradient length ($\delta = 6 \text{ ms}$) gives an estimation of minimal diffusion value of $2.5 \cdot 10^{-12} \text{ m}^2\text{s}^{-1}$. If we introduce this value in the Einstein Stokes relation, assuming we are in a water solution at $25 \text{ }^\circ\text{C}$ we obtain a r_h value of approximately 100 nm as the maximum theoretical size we could measure in our conditions.

New experimental data including r_h calculation for different size NPs has been added in the Supporting Information, sections VI.2-3 (pages 61-75). Also a comment regarding the maximum theoretical size that could be measured has been added in page 75.

- Mixtures of thiolated ligands are grafted on GNPs. Their ratio at the GNP surface is assumed to be the same as in solution. However, grafting of different molecules using thiolated ligands can difficultly be controlled. ^1H NMR spectroscopy should allow an easy characterization of the grafting densities of the different ligands on the NPs, which could influence the adsorption of proteins. TGA would also be a more robust method to determine the fraction of organic ligands compared to ICP-MS. **The characterization of the functionalized NPs should be improved.**

⇒ Direct analysis of the ligands on the NPs by NMR is complicated and biased in this case since the signal intensities of the ligands may be altered by the fact that they are grafted on the surface of a NP. Hence, we have analyzed the ratio of the recovered unbound ligand mixture after ultracentrifugation at 150.000 g. That analysis shows that the ratio is basically maintained and that we have an approximate ratio of 75/25, as desired (Figure S.I-15). To further probe this, we submitted NPs directly to elemental analyses and we confirmed again that the ligand ratio on the NPs is still 75/25 and was also consistent with the ICP data (Table S.I-1). In line with the improvement of the characterization of the NPs we have also added new measurements in section I as for instance, IFT and T_2 values measurement.

New characterization data can be found in section I.3 of the Supporting Information (pages 11-21)

- ^{19}F DOSY experiments are performed under spinning. Contradictory indications are given in the literature regarding the spinning or not of the samples during DOSY experiments. However, as reported in publications by the Gareth Morris group: “The problems with spinning are that it generates mechanical instabilities, and that unless scrupulous care is taken to synchronise the timing of gradient pulses with the sample rotation, small spatial inhomogeneities in the gradient field will lead to large signal disturbances. For these reasons sample spinning is not recommended for accurate diffusion measurements. »(see <http://dx.doi.org/10.1016/j.jmr.2014.12.006>). **As the differences in the diffusion coefficients measured in different sets of conditions are quite small, have these precautions be taken in order to ensure that experiments are run in the best conditions? Δ (“big delta”) should be mentioned as diffusion delay and δ (“little delta”) as the gradient length.**

⇒ We have changed big delta by diffusion delay and little delta by gradient length, as suggested by the referee. Regarding the rotation during the diffusion measurements, we have performed several measurements of the same sample with and without rotation and we have not observed major differences. With rotation, as we normally do, diffusion constant for NP-F/NH₂ in water was measured to be $(2.86 \pm 0.04) \cdot 10^{-11} \text{ m}^2/\text{s}$, and the same sample without rotation rendered a diffusion constant value of $(2.85 \pm 0.15) \cdot 10^{-11} \text{ m}^2/\text{s}$. Hence, we do not believe that rotation is a major issue in our system.

We have included a comment on this in page 27 of the Supporting Information. In addition we have also cited the reference mentioned by the referee as ref. 12 in our Supporting Information.

- Colloidal suspensions are difficult to stabilize in physiological conditions. Was the diffusion coefficient of the NP suspensions characterized in different ionic strength conditions in order to ensure that the - often weak - observed effects are not induced by a change in the IS of the suspension?

⇒ We measured NPs both in plain water and in PBS or HEPES 100 mM and the differences are very subtle, as shown in Figure S.V-5. In addition, all the points of each of the curves in figures S.IV-1-4 were measured in the same buffer to avoid changes due to variations in the ionic

strength or viscosity derived from the solution medium. We agree with the reviewer that colloidal stability is paramount. As well the PMA as the PEG coatings have been investigated by our group over the years and all characterization, for example stability in electrolytic solutions has been done (see ref. 14 of the manuscript).

We have added a statement about this in the third paragraph of the main manuscript (page 2).

- The measurement, based on ^{19}F NMR DOSY experiments, is smart but restricted to fluorinated ligands. To which extent could this layer of F atoms exposed to the solvent modify the NP/protein interactions (as suggested from Figure 2)? Has protein adsorption been characterized with fluorinated vs hydrogenated ligands using more conventional methods to highlight potential differences?

⇒ We varied the amount of fluorine in NP-F/COOH from 100 % of HS-PEG-F (NP-F) to 50 % (NP-F/COOH(50)) and those NPs were incubated with a high concentration of HSA (1.5 mM) and the behavior was very similar independently from the amount of fluorine on the NP. However, we did observe differences when we changed the functionalities of the accompanying ligand (from amino to carboxyl groups) with respect to NP-F after incubation with HSA or Fibrinogen. Apparently, both charge and hydrophobicity may play a major role in the interaction with proteins with PEGylated NPs, as we observed by comparing IFT and zeta potential data with diffusion measurements. New experimental data regarding this can be found in new section VI.1 in the Supporting Information (pages 56-61).

In any case, to avoid restrictions coming from the presence of either fluorine or PEG that could hamper the applicability of the methodology, we have also reported on polymer coated NPs that do not have a fluorinated external surface, that is NP-F/ NH_2 @PMA (Figure 2 in the main manuscript).

We have also added some discussion from literature (as in particular obtained on planar substrates). Data indicate that there is protein adsorption also on fluorinated surfaces (see ref. 18 in the manuscript), but that adsorption of certain proteins can be minimized (see ref. 19 in the manuscript).

- The developed method is compared to FCS and DLS to quantify in situ formation of a protein corona around NPs. However, these advantages are not clearly described and the limitations are not discussed:

1) *Sensitivity of the NMR technique*: the whole fraction of ^{19}F nuclei contained in a 2ml sample is requested to perform a measurement in reasonable time. How could this be extrapolated to particles dispersed in tissues or injected in animals, as the particles, distributed in these complex matrices, will experience different environments and diffusion coefficients (in addition to the fact that blood flow is necessary in a living organism) ?

⇒ The idea is to measure diffusion in the animal as a whole in the blood stream. Hence, the measurement has to be performed as the probe is injected and most of the sample is in the blood stream. In addition, to begin to work we could assume that diffusion in blood would be

substantially faster than in tissue, so it could be easily distinguished. But of course this is and hypothesis yet to be proved. We have added a more detailed discussion about the problems of such future measurements at the end of the main manuscript (pages 5-6).

2) *Dependence on the suspension characteristics*: sample viscosity is a crucial parameter in the Einstein-Stokes relation. Even in a media as plasma, TFA needs to be used as internal reference in order to determine the viscosity of the sample – and the 10% precision obtained is an important source of uncertainty on the calculated r_h value. Should TFA also be injected into animals when performing in vivo experiments? The variations (decrease) obtained for the r_h of NP-F/CO₂H and NP-F/NH₂ in blood/plasma vs water highlight this problem. Similarly, the decreased r_h for the NP-F/NH₂ in the presence of HSA should be commented. **If the effect of the physico-chemical properties of the suspension (ionic strength, pH,...), or the presence of plasma components that lead to a shrinkage of the PEG layer, cannot be distinguished from the interaction with proteins, no information on the protein corona formation can be extracted from the measurements!**

⇒ We agree. As an approximation we could begin by using literature data on blood viscosity. Injecting TFA does not seem feasible for the viability of the animal. But we could extract blood and add TFA to measure viscosity in an NMR tube, since this measurement is very short and blood should stay in one phase in the first 20 minutes approximately, as we have observed for human blood. The measurements have been performed at physiological pH when natural adsorption has been studied, hence we can rule out pH as a variable. As explained before, the ionic strength has also been studied by measuring our samples in water, PBS or HEPES and we did not observe significant differences in size.

In conclusion, if the introduced methodology is very elegant and allows the measurement of diffusion coefficient of NPs in complex media without interference of the protein signals (or of those of other analytes present in blood samples), its limitations and weaknesses are not sufficiently discussed and the obtained information as the potential in-vivo application seems largely overestimated.

⇒ Yes, we agree with the reviewer. We have tried to reduce our claims and to add a better discussion about the application of our method.

REVIEWERS' COMMENTS:

Reviewer #1 (Remarks to the Author):

The modified version of Carril et al.'s manuscript is a work of major importance to the field of nanobiotechnology. It is the first time that a method with the potential to dynamically interrogate the formation of a biomolecular corona in an in vivo environment has been described. This method will (hopefully) provide new insights into the complex fate of nanomaterials within the body. The authors have thoroughly addressed all of the concerns I raised in my original review. While they have not demonstrated the application of their technique to an actual in vivo case, they have set a solid foundation for a follow-up study to do just that. I can now confidently recommend the manuscript for publication in Nature Communication in its current form.

Reviewer #2 (Remarks to the Author):

Carril et al. has improved an already impressiv article. They have addressed all of my questions and provided clear answers or modifications to the article (I stand corrected for not reading the SI in the reference). Caril et al. work present a new tool to investigate the formation of protein coronas around nanoparticles and will be of interest for readers from many different disciplines. I recommend this article for publication.

M. Lundqvist

Reviewer #3 (Remarks to the Author):

Review on the revised manuscript entitled « In situ detection of the protein corona in complex environments » submitted by Monica Carril, Daniel Padro, Pablo del Pino, Carolina Carrillo-Carrion, Marta Gallego and Wolfgang J. Parak for publication in Nature Communications.

As already mentioned for the previous version of the submitted manuscript, the strategy introduced by Carril et al. is particularly clever. The efforts and amount of work achieved by the authors to answer to all the comments of the referees is also particularly impressive. For these reasons I would recommend the publication of the manuscript despite the fact that further optimization is requested to reveal the full potential of this methodology to study the interaction between nanoparticles and biological (macro)molecules in complex media, and ideally in vivo, by ^{19}F DOSY NMR measurements.

The following comments would however have to be addressed :

- One major drawback of the methodology seems to be linked to the « softness » of the PEG layer that can be compressed or deformed in some experimental conditions. This « shrinkage » of the organic layer makes the interpretation of the modifications of the hydrodynamic radius extremely tricky : indeed, the observation that the rh remains unchanged in the absence and in presence of different proteins do not imply that there is no interaction: it is indeed possible to imagine that an increase of rh linked to the adsorption of a protein could be compensated by the shrinking of the PEG layer, leaving the rh unchanged. In these conditions only two cases can be clearly distinguished : (i) the adsorption of proteins leading to a small increase of rh and (ii) the aggregation of the particles leading to a drastic change of their rh. This drawback should be discussed in the conclusions where for the moment protein adsorption is only associated to an increase in hydrodynamic radius.

- The upper limit of 100 nm for the maximum measurable size of particles seems largely overestimated as it is doubtful that ligands anchored on such large particles will present a T2 that is large enough to allow 0,6 s of diffusion delay in the DOSY experiments. The question of the observability of the ligands anchored on large particles (even if a difficult question as not only linked to the rotational correlation time of the ensemble but also to the intrinsic mobility of the ligands) should also be considered.
- Regarding a comment on the initial submission, linked to the ratios of PEG ligands at the surface of the particles (that the ratio at the surface might be different than the ratio in solution), the fact that the ratio of HS-PEG-F and HS-PEG-COOH in the first supernatant remains close to the initial ratio in solution is not a proof that these two molecules have been grafted in the right ratio. Indeed, if only a small fraction of the total PEGs added is grafted, the quantity retrieved from the solution (i .e. the first supernatant) might not be sufficient to be highlighted by NMR spectroscopy, or could be reflected by the small difference observed in the ratio before (75/25) and after (72/28) grafting. Grafting densities of 1 PEG/nm² are often found in the literature for PEG larger than 1kDa and this would indeed lead to a loss of a very small fraction of the total PEG in the first supernatant in the conditions used (less than 10%).

As already mentioned, I would recommend publication after these minor modifications.

Reviewer #3 (Remarks to the Author):

Review on the revised manuscript entitled In situ detection of the protein corona in complex environments submitted by Monica Carril, Daniel Padro, Pablo del Pino, Carolina Carrillo-Carrion, Marta Gallego and Wolfgang J. Parak for publication in Nature Communications.

As already mentioned for the previous version of the submitted manuscript, the strategy introduced by Carril et al. is particularly clever. The efforts and amount of work achieved by the authors to answer to all the comments of the referees is also particularly impressive. For these reasons I would recommend the publication of the manuscript despite the fact that further optimization is requested to reveal the full potential of this methodology to study the interaction between nanoparticles and biological (macro)molecules in complex media, and ideally in vivo, by ^{19}F DOSY NMR measurements.

The following comments would however have to be addressed :

- One major drawback of the methodology seems to be linked to the softness of the PEG layer that can be compressed or deformed in some experimental conditions. This shrinkage of the organic layer makes the interpretation of the modifications of the hydrodynamic radius extremely tricky : indeed, the observation that the rh remains unchanged in the absence and in presence of different proteins do not imply that there is no interaction: it is indeed possible to imagine that an increase of rh linked to the adsorption of a protein could be compensated by the shrinking of the PEG layer, leaving the rh unchanged. In these conditions only two cases can be clearly distinguished : (i) the adsorption of proteins leading to a small increase of rh and (ii) the aggregation of the particles leading to a drastic change of their rh. This drawback should be discussed in the conclusions where for the moment protein adsorption is only associated to an increase in hydrodynamic radius.

We completely agree with the reviewer that the presented methodology does not allow us to obtain information on what is actually happening on the nanoparticle surface beyond measuring the changes in hydrodynamic size. Hence, as the reviewer points out, the fact that there is no size change does not imply necessarily that there is no interaction with proteins. Indeed, we agree that our PEGylated nanoparticles NP-F, NP-F/COOH or NP-F/NH₂ may not be the best system for studying the formation of protein corona but they worked very nicely for the setting up of the measuring conditions and as a proof of concept. In this regard, we believe that the PMA coated NPs, NP-F/NH₂@PMA and [NP-F/NH₂@PMA]*2 are better systems since we did not observe size compression in any case, probably due to the protective PMA layer, and we only observed size increase in the presence of proteins that fitted the Hill model. A sentence related to this comment has been included in the conclusions as requested, and highlighted in yellow.

- The upper limit of 100 nm for the maximum measurable size of particles seems largely overestimated as it is doubtful that ligands anchored on such large particles will present a T₂ that is large enough to allow 0,6 s of diffusion delay in the DOSY experiments. The question of the observability of the ligands anchored on large particles (even if a difficult question as not only linked to the rotational correlation time of the ensemble but also to the intrinsic mobility of the ligands) should also be considered.

We agree with the reviewer that the value presented as the upper size limit is indeed an optimistic value. As the reviewer mentions, the key point of our estimation is to know how ^{19}F NMR signal behaves when the fluorinated ligand is anchored to a nanoparticle. To the best of

our knowledge, currently it is not possible to accurately predict this behavior and the particular design of the fluorinated nanomaterial to be used will play a major role in the detection of the ^{19}F NMR signal and its T_2 value. In this scenario, the possibility of measuring the diffusion constant by ^{19}F NMR will be mainly limited by 3 parameters: concentration of fluorine in the sample, T_2 value of fluorine atoms and the diffusion coefficient of the nanomaterial. In the fluorinated nanoparticles presented in this manuscript we have found that the T_2 values of fluorine atoms are long enough not to play an important limitation in our measurements. We completely agree that the use of the same fluorinated ligands on bigger nanoparticles will most likely lead to a decrease in the T_2 value and will affect the fluorine concentration. These two factors will certainly limit the maximum size measurable. However, nanoparticle associations such as that described herein of $[\text{NP-F}/\text{NH}_2@\text{PMA}]^*2$ allowed us to double the size of our initial nanomaterial without substantial decrease in the T_2 value, as described in detail in Supplementary Figure 62. Hence, the value presented as the upper limit was the only value that could be estimated with the known limitations (mainly presented by hardware) and assuming that T_2 and concentration would not be a limitation. Nonetheless, we realize that the design of such a big probe with long relaxation times is highly challenging at this stage. We have included a paragraph in the supplementary methods and in the manuscript stating clearly the limitations in the size calculation and have been highlighted in yellow.

- Regarding a comment on the initial submission, linked to the ratios of PEG ligands at the surface of the particles (that the ratio at the surface might be different than the ratio in solution), the fact that the ratio of HS-PEG-F and HS-PEG-COOH in the first supernatant remains close to the initial ratio in solution is not a proof that these two molecules have been grafted in the right ratio. Indeed, if only a small fraction of the total PEGs added is grafted, the quantity retrieved from the solution (i.e. the first supernatant) might not be sufficient to be highlighted by NMR spectroscopy, or could be reflected by the small difference observed in the ratio before (75/25) and after (72/28) grafting. Grafting densities of 1 PEG/nm² are often found in the literature for PEG larger than 1kDa and this would indeed lead to a loss of a very small fraction of the total PEG in the first supernatant in the conditions used (less than 10%).

The reviewer is right and determining the ratio of ligands based only on the ratio in the NMR of the supernatants is not totally reliable, but it works only as a rough orientation. For this reason we also performed elemental analysis directly on the nanoparticles to confirm that the ratios obtained by NMR of the recovered unbound ligand were correct, as it was already included in the supplementary file in the previously revised version. In addition, since our ICP-MS measurements have been performed on dried NP samples of known weight, it has been possible to estimate the number of PEG ligands per nm² and for our case it is 5 PEG/nm², we have included this calculation in the ICP-MS section of the revised supplementary information. Nonetheless, for our purposes we were not very concerned with the exact ratio, we only wanted a reasonable fraction of carboxyl groups to have a few anchoring points for protein linking with EDC/NHS chemistry.

As already mentioned, I would recommend publication after these minor modifications.